# YAP and ECM Stiffness: Key Drivers of Adipocyte Differentiation and Lipid Accumulation

**DOI:** 10.3390/cells13221905

**Published:** 2024-11-18

**Authors:** Da-Long Dong, Guang-Zhen Jin

**Affiliations:** 1Institute of Tissue Regeneration Engineering (ITREN), Dankook University, Cheonan 31116, Republic of Korea; dongdalong@dankook.ac.kr; 2Department of Nanobiomedical Science and BK21 PLUS NBM Global Research Center for Regenerative Medicine, Dankook University, Cheonan 31116, Republic of Korea; 3Department of Biomaterials Science, College of Dentistry, Dankook University, Cheonan 31116, Republic of Korea

**Keywords:** YAP signaling, ECM stiffness, adipocyte differentiation, hydrogels, mechanotransduction, lipid accumulation, cellular mechanobiology, fat tissue engineering

## Abstract

ECM stiffness significantly influences the differentiation of adipose-derived stem cells (ADSCs), with YAP—a key transcription factor in the Hippo signaling pathway—playing a pivotal role. This study investigates the effects of ECM stiffness on ADSC differentiation and its relationship with YAP signaling. Various hydrogel concentrations were employed to simulate different levels of ECM stiffness, and their impact on ADSC differentiation was assessed through material properties, adipocyte-specific gene expression, lipid droplet staining, YAP localization, and protein levels. Our results demonstrated that increasing hydrogel stiffness enhanced adipocyte differentiation in a gradient manner. Notably, inhibiting YAP signaling further increased lipid droplet accumulation, suggesting that ECM stiffness influences adipogenesis by modulating YAP signaling and its cytoplasmic phosphorylation. This study elucidates the molecular mechanisms underlying ECM stiffness-dependent lipid deposition, highlighting YAP’s regulatory role in adipogenesis. These findings provide valuable insights into the regulation of cell differentiation and have important implications for tissue engineering and obesity treatment strategies.

## 1. Introduction

The extracellular matrix (ECM) significantly influences adipocyte differentiation and plasticity, affecting both their maturation and potential for dedifferentiation [1,2]_._ Recent studies have revealed the complex interactions between ECM components and adipocytes, highlighting the mechanisms regulating the adipocyte differentiation process. For example, ECM proteins like fibronectin and type I collagen can substantially improve mature adipocyte conversion into dedifferentiated fat cells (DFATs) with an increase in their stem cell-like properties [2]. A similar test using the autografts of decellularized extracellular matrix (d-ECM) successfully induced adipogenic differentiation of ADSCs via the ERK1/2-PPARγ signaling pathway, which is crucial for adipocyte maturation [3,4].

Early studies on mechanotransduction predominantly used 2D culture models to investigate how substrate stiffness affects cell migration, proliferation, malignancy, and differentiation [5,6,7,8,9]. These studies elaborated on the concept of mechanotransduction, demonstrating that their findings are applicable in vivo. However, in most cases, a 3D cultural environment better presents behavior that is relevant biologically [10,11]. A 3D microenvironment can sustain the chondrogenic phenotype [12], distinguish between normal mammary epithelial cells and breast cancer cells, allowing only the normal cells to form growth-arrested acinar structures [13], enhance biological activities of cells adhering to a 3D matrix as observed in vivo [11], improve the pluripotency in human embryonic stem cells [14], and regulate the angiogenic capacity of tumor cells [15]. Notably, recent evidence indicates that mechanotransduction in 3D is fundamentally different from that in 2D [12,13,14,15].

ECM stiffness and hydrogels regulate YAP activity. A key driver of mechanotransduction, YAP can influence stem cell fate by perceiving and responding to mechanical signals conveyed by ECM stiffness. In 2D cultures, YAP nuclear localization varies with substrate stiffness. A stiffer ECM promotes YAP translocation into the nucleus, thereby facilitating cell differentiation [16]. However, the 3D culture microenvironment is essential for eliciting in vivo-like behaviors and may differentially affect YAP compared to 2D systems. For example, hydrogels of tunable stiffness can drive lineage-specific differentiation of stem cells through regulating the mechanosensing ability of YAP [17]. In 3D cultures, spatially directing YAP nuclear translocation via hydrogel degradation patterns further mediates ADSCs differentiation [18].

YAP interacts with mechanical signals: A stiff ECM can activate YAP and promote changes in glycolysis and mitochondrial dynamics that are vital for stem cell differentiation [19]. In engineered hydrogels, an increase in stiffness significantly enhances YAP signaling, thereby promoting cell remodeling and differentiation processes [20]. This mechanosensing capability of YAP enables it to integrate physical signals from the ECM and influence the differentiation pathways of ADSCs by regulating downstream gene expression [21].

Type I collagen is the most abundant structural protein found in the ECM of humans and mammals. We utilized collagen hydrogels with adjustable matrix stiffness to replicate the in vivo 3D microenvironment and systematically examine the impact of hydrogel matrix stiffness on the differentiation of ADSCs. Our work explored how adipose stem cells perceive their surrounding microenvironment in a 3D hydrogel and how this influences the process of adipocyte differentiation. Material phenotype analyses, gene expression, YAP nuclear translocation, Western blotting, and adipogenesis-related fluorescence quantification were conducted to clarify the role of YAP in regulating adipocyte differentiation.

## 2. Materials and Methods

### 2.1. Cell Culture

SD rats were obtained from JSBIO, Korea, and housed at Dankook University’s animal facility with ad libitum access to food and water under a 12 h light/dark cycle. All animal experiments adhered to the ARRIVE guidelines and were approved by the Dankook University Institutional Animal Care and Use Committee, following relevant regulations. Inguinal fat tissue from 4-week-old Sprague-Dawley rats was aseptically harvested using sterile surgical scissors. The tissue was minced and digested with 0.2% collagenase type I by continuous shaking at 37 °C for 30 min. The digested mixture was filtered through a 40-micron cell strainer to eliminate undigested tissue fragments. The cells were cultured in high-glucose DMEM with 10% fetal bovine serum (FBS) and 1% (*v*/*v*) penicillin/streptomycin (100 U/mL penicillin, 100 µg/mL streptomycin; Sigma Aldrich, UK). When the cell confluence reached 80%, cell cryopreservation was performed. Passage 4 (P4) cells were used for the experiments.

### 2.2. Hydrogel Preparation

A hydrogel matrix consisting of rat tail collagen I (Corning Incorporated, Product batch number: 354236, Corning, NY, USA) at 3.47 mg/mL was used in this study. Hydrogels were prepared at concentrations of 0.5, 1, and 2 mg/mL according to the manufacturer’s instructions and experimental criteria. We diluted Dulbecco’s modified Eagle’s medium (DMEM High Glucose, Catalog number: LM 001-05, Manufacturer: BioInd, Kibbutz Beit Haemek, Israel) and collagen in the specified volume ratio. We adjusted the pH to 7.2 using 1 N sodium hydroxide solution. We dispensed 100 µL of the hydrogel mixture into silicone molds with a dimension of 0.9 cm diameter and 0.2 cm thickness. We then incubated the molds at 37 °C for 20 min to allow gelation. After gelation, we carefully demolded and added differentiation medium, as detailed in Appendix A.

### 2.3. Stress and Elastic Modulus Testing of Adipose Tissue and Hydrogels

We selected abdominal fat tissue from 6-week-old rats, ensuring it was harvested as intact as possible. The fat tissue was strip-shaped, becoming progressively thicker from left to right. The tissue was divided into three sections, named APT, MPT, and PPT, for testing. The same compression testing method used for the hydrogels was applied, with a maximum compression of 60% every 200 s.

The cell-free hydrogels were soaked in distilled water until they reached swelling equilibrium, after which their mechanical properties were evaluated using a dynamic mechanical analyzer. The compressive modulus was measured by applying a 60% compressive strain at a strain rate of 200 s^−1^ at room temperature, utilizing the CellScale MicroTester mechanical testing instrument (CellScale MicroTester G2, CellScale MicroTester mechanical testing instrument, CellScale, Waterloo, ON, Canada). Each test group consisted of five parallel samples, and the average of the results was calculated.
$$\sigma =\frac{F}{A}\;\;,\;\;\;\;E=\frac{\sigma }{\varepsilon }$$

σ: Stress in units of pascals (Pa)F: The force applied to the material, in units of newtons (N)A: The material’s cross-sectional area, measured in square meters (m^2^)E: Young’s modulus is measured in pascals (Pa).ϵ: strain (%)

### 2.4. ADSCs Differentiate into Adipocytes in Hydrogel

For the adipogenic differentiation process, the cells were thoroughly mixed with each collagen hydrogel to reach a final concentration of 1.5 × 10^5^ cells/mL. The cell-seeded hydrogels were cultured in DMEM high-glucose medium supplemented with 10% FBS and 1% (*v*/*v*) PS. The differentiation medium included ITS1X (ITS 1X, BD Biosciences, Oxford, UK), 10 nM dexamethasone (Dexamethasone, 10 nM, Sigma Aldrich, Dorset, UK), and 283.897 μM ascorbic acid. The hydrogels were maintained at 37 °C with 5% CO_2_ in a cell culture incubator, and the differentiation medium was refreshed every other day during the 9-day culture period.

### 2.5. Evaluation of Compatibility Between Adipose Stem Cells and Collagen Hydrogel

Using the Live/Dead^®^ Cell Viability Assay Kit (No.1) (R37601, Life Technologies, Carlsbad, CA, USA), the viability of ADSCs embedded in hydrogels was evaluated after 24 h. The assay was conducted based on the instructions provided by the manufacturer. The hydrogels were incubated with dye in a 4-well plate for 15 min at room temperature in the dark. Subsequently, the hydrogels were washed with phosphate-buffered saline (PBS) and finally immersed in PBS for observation and image acquisition under a microscope.

### 2.6. Influence of Hydrogel Matrix Stiffness on Adipocyte Proliferation

To assess the impact of matrix stiffness on adipocyte proliferation within 3D hydrogels, four hydrogels from each concentration group were randomly chosen and placed in 1.5 mL centrifuge tubes. We added 100 µL of 0.2% collagenase I solution to each tube and incubated them in a 37 °C water bath for 30 min. Post-digestion, the samples were centrifuged at 2000 rpm for 5 min, and the supernatant was discarded. The Cell Counting Kit-8 (Dojindo Laboratories, Kumamoto, Japan, Code: CK04, Lot: WY612) reagent was diluted 1:10 with serum-free DMEM (Welgene, Gyeongsan, Republic of Korea, Catalog number: LM 001-05, Lot: LM01243405) medium. Subsequently, 100 µL of CCK-8 mixture was added to each sample, and the samples were incubated at 37 °C for two hours. Finally, we measured the absorbance at 450 nm with a microplate reader to determine the peak value of cell proliferation and perform statistical analysis.

### 2.7. Quantification of Lipid Accumulation in ADSCs Using BODIPY Staining

To quantify intracellular lipid accumulation, we used BODIPY 493/503 (Cat# HY-W090090, MedChemExpress, USA) to fluorescently label lipids. The cultured hydrogels were first fixed for 10 min in 4% paraformaldehyde (PFA). The samples were washed thrice with PBS for 5 min each and then permeabilized with 0.2% Triton X-100 for 15 min at room temperature. The samples were washed three times with PBS for five minutes per wash. The cells were stained with 5μM BODIPY for 15 min and then washed three times with PBS. Cell nuclei were stained with 2 ng/mL DAPI for 15 min, followed by a PBS wash. Fluorescence microscopy was employed for imaging, and ImageJ 1.52P software quantified intracellular green fluorescence indicative of lipid droplets. Statistical analyses were conducted using GraphPad Prism 8.0.2 software.

### 2.8. Real-Time PCR Quantitative Analysis

Following digestion with 0.2% type I collagenase for 30 min, the cells were centrifuged, and the total RNA was isolated using TRIzol reagent (ThermoFisher). cDNA was generated using Bio-Rad’s iScript cDNA Synthesis Kit (IScript cDNA Synthesis Kit, Bio-Rad, Hercules, CA, USA). A qRT-PCR program was conducted using the StepOne Plus Real-Time PCR System (Applied Biosystems, Foster City, CA, USA) and the SensiMix SYBR Hi-ROX kit (Bioline, London, UK, Catalog number: QT-605-05). Table 1 provides a list of primers used.

### 2.9. Measurement of Changes in Hydrogel Diameter After Cell Seeding

After the cells were inoculated into the hydrogel, the hydrogel was removed from the silicone mold after solidification. A standard ruler was used as a scale for the size of the hydrogel (see Appendix A), and the diameter change of the cell hydrogel was measured at 6, 12, 24, and 48 h after inoculation. The contraction data of the hydrogel at different time periods were recorded in detail. Five data sets per concentration were measured and analyzed statistically using GraphPad Prism 8.0.2 software.

### 2.10. YAP Immunofluorescence Staining

After 9 days, the cells were fixed in 4% PFA for 15 min, washed three times in PBS for 5 min each, permeabilized with 0.2%Triton X-100 for 10 min, and washed again three times in PBS for 5 min each. Next, we blocked non-specific binding with 5% BSA for 60 min. The YAP antibody (Santa Cruz Biotechnology, Catalog number: sc-101199, Dallas, TX, USA) was diluted 1:200 and incubated at 4 °C overnight. The following day, we performed three 10 min PBS washes to eliminate unbound primary antibodies. Fluorescently labeled secondary antibodies were diluted 1:1000 and incubated in the dark for 1 h. Nuclei were stained with a DAPI solution of 1 µg/mL for 5–10 min at room temperature. Finally, we washed three times with PBS for 5 min each. Images were taken with a confocal microscope for the observation of YAP localization. While the nuclei were stained blue by DAPI, YAP was stained in green by Alexa Fluor 488. Images were merged for further analysis.

### 2.11. Western Blot

The cells were extracted from the hydrogels according to previously established methods. The cells were lysed on ice for 30 min using protease and phosphatase inhibitors (HaltTM Protease and Phosphatase Inhibitor Cohcktail, 100X, Thermo Scientific, Waltham, MA, USA; EBA-78440, EBA-78440, Elpis Biotech, Daejeon, Republic of Korea). The protein concentrations in the supernatant were measured using the Pierce BCA Protein Assay Kit (Thermo Scientific, Waltham, MA, USA) after centrifugation at 10,000 rpm for 10 min at 4 °C, according to the manufacturer’s instructions. Samples were heated at 100 °C for 10 min to denature them before separation via SDS-PAGE (Gel Solution, Bio-Rad, Hercules, CA, USA). Protein electrophoresis was followed by a transfer to PVDF membranes. The membranes were blocked with BSA (SolMate BSA Grade IY, GeneAll, Seoul, Republic of Korea) for 60 min and then incubated overnight at 4 °C with primary antibodies against phospho-YAP and GAPDH. Following three TBST washes the next day, the membrane was incubated with HRP-conjugated antibody for 1 h. The protein signals were captured using the LAS4000 mini protein imaging system (GE Healthcare, Uppsala, Sweden.) with Supersignal West Pico and Supersignal West Pico PLUS Chemiluminescent Substrates (Thermo Scientific). Quantitative analysis was conducted using ImageJ 1.52P software. Table 2 provides a list of primers used.

### 2.12. Porosity Analysis of Freeze-Dried Hydrogels

After preparing the cell-free hydrogel samples, they were subjected to rapid freezing in liquid nitrogen followed by freeze-drying. Platinum was sputter-coated onto the freeze-dried hydrogels using a Japanese sputter coater, the Eiko IB-3 (Eiko Engineering, Tokyo, Japan). A scanning electron microscope (JEOL-SEM 3000, JEOL Ltd., Akishima, Tokyo, Japan) operated at 5 kV acceleration voltage was then used to determine the porosity of the freeze-dried hydrogels. The physical support provided by the porous structure was evaluated, and the influence of cross-sectional pore size on cell differentiation was investigated. Finally, quantitative analysis of pore size distribution was conducted using Nano Measurer 1.2 software.

### 2.13. Effects of YAP Inhibition on Lipid Droplets in Hydrogels

This experiment aimed to investigate the effect of verteporfin (VP), an inhibitor of the YAP signaling pathway, on lipid droplet formation in hydrogels. On day 7 of differentiation, the hydrogels were treated with 1 µM VP. Two experimental groups were formed: a control group and a YAP inhibition group with an inhibition time of 48 h. Four randomly selected hydrogels were used in each experimental group. These experiments involved the use of BODIPY staining to assess lipid droplets in normal and suppressed groups.

### 2.14. Statistical Analysis

Statistical analysis was conducted using GraphPad Prism 8.0.2 software. Data are presented as the mean ± standard deviation. One-way ANOVA and multiple comparisons, as well as unpaired *t*-tests, were employed to assess statistical significance among the re-search groups. Western blots and BODIPY staining images were quantitatively analyzed using ImageJ 1.52P software. *p* < 0.05 was considered to be statistically significant.

## 3. Results

### 3.1. From Tissue to Hydrogel: Constructing an In Vitro Model for Accurate Simulation of the Microenvironment for Adipocyte Differentiation

To accurately simulate the microenvironment of adipocyte differentiation in vitro, we first extracted samples from rat inguinal adipose tissue and measured the stress and elastic modulus at different sites. These tests established benchmarks for the mechanical properties of real adipose tissue, including stress and elastic modulus, to guide the development of the in vitro model.

The rat inguinal adipose tissue was strip-shaped, with the thickness ranging from thin to thick. We further divided the entire adipose tissue into three parts: the anterior part of the adipose tissue (ATP), the middle part of the adipose tissue (MPT), and the posterior part of the adipose tissue (PPT) (Figure 1A); and we then tested stress and elastic modulus of each part. The results demonstrated a gradual increase in both stress and the elastic modulus from the ATP to the PPT (Figure 1B). For the samples, the compressive stress values were measured as 294.6 ± 41 Pa for ATP, 606.3 ± 90.7 Pa for MPT, and 1013.6 ± 89 Pa for PPT. Under a strain condition of 60%, the elastic modulus was determined to be 589.2 ± 81.9 Pa for ATP, 1212.7 ± 181.6 Pa for MPT, and 2027.2 ± 178 Pa for PPT (Figure 1A–C). These differences may be attributed to variations in thickness and cell density within the adipose tissue specimens. Thicker sections contain more adipocytes, leading to greater cell–cell mechanical interactions and a higher elastic modulus. Thinner sections, therefore, are likely to contain more connective tissue or blood vessels, which have a lower elastic modulus.

With these data as a reference, we sequentially formulated hydrogels at concentrations of 0.5 mg/mL, 1 mg/mL, and 2 mg/mL to replicate different mechanical environments. The corresponding hydrogel stress and elastic modulus were 344 Pa, 982 Pa, and 1220 Pa, as well as 301.8 ± 43.9 Pa, 1561 ± 98.6 Pa, and 2250.58 ± 89.62 Pa, respectively, closely approximating the mechanical properties of in vivo adipose tissue (Figure 1D,E). This design established an in vitro environment that mimics in vivo conditions, allowing for the study of the impact of hydrogel mechanical properties on adipocyte differentiation.

The microenvironment of adipose tissue, along with the stiffness of hydrogels, plays a crucial role in supporting cellular processes such as proliferation, differentiation, and migration. Conversely, the exchange of substances between cells and their immediate environment is equally vital for cell survival, with the pore structure of hydrogels playing a key role in this process. Pore size directly affects the uptake of external nutrients by cells and regulates cellular behavior at both physical and chemical levels. Therefore, the pore structure of hydrogels can significantly mimic an in vivo environment, thereby influencing cell differentiation.

The process by which cells perceive and respond to these mechanical signals is called mechanotransduction. Mechanotransduction converts forces into biochemical signals, altering cell behavior by activating intracellular signaling pathways, regulating gene expression, and modifying cell shape and cytoskeletal organization [19,22,23]. To further investigate the pore characteristics of hydrogels at different concentrations, the analysis of freeze-dried hydrogels was conducted using SEM (JEOL-SEM 3000, JEOL Ltd, Akishima, Tokyo, Japan). Results are presented in (Figure 1F,G). The results indicated that pore size increased with concentration. Hydrogels containing 0.5 mg/mL, 1 mg/mL, and 2 mg/mL exhibited pore sizes of 1.5–3 µm, 2–4 µm, and 2.5–5 µm, respectively. The pores in 0.5 mg/mL hydrogel were mainly concentrated within the range of 1.8 and 2.5 µm. For the 1 mg/mL hydrogel, pore sizes ranged from 2 to 3.5 µm, while the majority of pores in the 2 mg/mL hydrogel were between 2.8 and 4.7 µm.

The larger pores in the 2 mg/mL hydrogel, ranging from 2.5 to 5 µm, facilitate greater nutrient exchange between the cells and their environment, while also providing sufficient space for cell migration and growth. Similarly, such large pores enhance the capability of mechanical sensing of cells within the matrix, thereby improving cell–hydrogel interactions. The previous literature has demonstrated that cells can modulate their differentiation processes by sensing the mechanical properties and pore structure of the matrix [19,24]. Larger pores in the 2 mg/mL hydrogel may create a more favorable microenvironment for differentiation. In comparison to the lower concentration hydrogels (0.5 and 1 mg/mL), the 2 mg/mL hydrogel exhibits higher compressive stress and elastic modulus, along with a larger pore structure, which may enhance nutrient uptake and facilitate the reception of differentiation signals. This suggests that pore size is related to cell differentiation, providing 2 mg/mL hydrogels with a distinct advantage in promoting adipocyte differentiation.

### 3.2. Fluorescent Staining of Lipid Droplets in 3D Hydrogels

In a 3D environment, varying concentrations of collagen hydrogel significantly influenced the differentiation ability of ADSCs into adipocytes. This observation was confirmed by BODIPY staining after 9 days of differentiation, as shown in (Figure 2A,B). Compared to the other concentration groups, a greater number of green fluorescent lipid droplets were observed in the 2 mg/mL hydrogel. The fluorescence intensity of lipid droplets in the 2 mg/mL hydrogel increased by 30.44% compared to the 0.5 mg/mL hydrogel, promoting lipid accumulation and facilitating the differentiation of ADSCs into mature adipocytes. Additionally, when compared to the 1 mg/mL hydrogel, the fluorescence intensity of lipid droplets in the 2 mg/mL hydrogel increased by 17.74%. BODIPY staining demonstrated that the 2 mg/mL hydrogel significantly enhanced lipid droplet accumulation and adipocyte differentiation, indicating a more favorable microenvironment for the development of the adipocyte phenotype. Under 3D culture conditions, the 2 mg/mL collagen hydrogel is particularly suitable for promoting the adipogenic differentiation of ADSCs into mature adipocytes. The 3D multilayer fluorescent images illustrate the differentiation of mature adipocytes (see Appendix A for details). These observations suggest that the 2 mg/mL concentration of collagen hydrogel provided the optimal scaffold structure and biological microenvironment for promoting the adipogenic differentiation of ADSCs.

### 3.3. qRT-PCR Analysis of Adipocyte Marker

The stiffness of the matrix differentially affects specific adipogenic genes. qRT-PCR analysis revealed that LPL, ADIPOQ, PPARα, and FABP4 were differentially expressed during adipocyte differentiation. Figure 2C illustrates a significant 1.65-fold increase in ADIPOQ gene expression in the 2 mg/mL hydrogel compared to the 0.5 mg/mL concentration. In the 2 mg/mL hydrogel, the expression levels of FABP4, LPL, and PPARG genes increased by 18.17-fold, 16.65-fold, and 26.366-fold, respectively, compared to the 0.5 mg/mL concentration. The expression levels of LPL, ADIPOQ, PPARG, and FABP4 genes in ADSCs were significantly elevated in a 2 mg/mL collagen hydrogel. This suggests that this concentration creates a favorable microenvironment for adipocyte differentiation and enhances the transcription of these adipocyte-related genes.

### 3.4. Cytotoxicity Test

Cell viability and survival status were assayed using a live/dead cell staining kit. The staining was at room temperature for 15 min, after which the cells were washed again with PBS to remove extra dye. The stained hydrogels were then observed under a confocal microscope, and the recorded cell morphology and distribution are shown in Figure 2D. Through the experimental results, it can be observed that there was no obvious cell death in the hydrogel after 24 h of culture. Figure 2G shows our quantitative analysis of live cell counts. This indicates that the hydrogel culture conditions positively affected cell survival and activity. Furthermore, live cells were uniformly distributed within the gel and showed no morphological changes characteristic of apoptosis or necrosis.

### 3.5. Cell-Mediated Shrinkage of Hydrogels

Cells interact with hydrogels through mechanical and chemical signals to regulate their morphology and function. Hydrogels mimic the extracellular matrix (ECM), providing cells with a scaffold on which they can exert traction, leading to compaction and contraction of the hydrogel. These force interactions are co-regulated by mechanical and chemical signaling pathways between cells and hydrogels. As shown in Figure 2E, our study included hydrogel contraction experiments to evaluate the behavior of adipocytes within the hydrogel matrix. These experiments were conducted at different time points (6 h, 12 h, 24 h, and 48 h) to investigate the contraction behavior of adipocytes cultured in hydrogels of different concentrations. The hydrogels were placed under controlled conditions, and their contraction rates were measured at regular intervals. Our research results demonstrate varying degrees of hydrogel contraction. After 48 h of culture and contraction, the hydrogels exhibited significant size reductions compared to the initial size of 0.8 cm. The 0.5 mg/mL hydrogel shrank by 87.5%, the 1 mg/mL hydrogel shrank by 75%, and the 2 mg/mL hydrogel shrank by 55%. The concentration of the hydrogel significantly influenced adipocyte contraction behavior. When the same number of cells were embedded in hydrogels of different concentrations, their perception of the microenvironment was influenced by changes in matrix stiffness, leading to variations in the contraction of the hydrogels at different concentrations. 

### 3.6. Cell Viability and Proliferation

Biocompatibility is a crucial factor to prioritize in the application of biomaterials. In our experiment (Figure 2F), we embedded ADSCs in hydrogels of varying concentrations and assessed their survival and proliferation at 2, 4, 6, and 8 days. The results showed that cells maintained a high survival rate in hydrogels of all concentrations, with a significant trend of proliferation over time. Notably, in the 2 mg/mL concentration of hydrogel, cell proliferation exhibited a gradient increase with time. As the concentration of the hydrogel increased, there was also a significant gradient change in both the proliferation rate and distribution of the cells. This indicates that hydrogel concentration not only affects the survival of ADSCs but also influences their proliferation behavior to some extent. These findings highlight the significance of hydrogels as cell culture matrices, emphasizing the necessity for good biocompatibility. Furthermore, the physicochemical properties of hydrogels should be optimized based on specific application requirements to effectively regulate cell behavior.

### 3.7. Modulation of YAP Function and Localization by Hydrogel Stiffness

Extracellular matrix stiffness is a key mechanical factor in regulating YAP activity [25]. Dupont et al. found that increased matrix stiffness promotes the nuclear localization and activity of YAP [26]. However, our results appear to contradict this finding, possibly reflecting the unique response of adipocytes to mechanical stimuli. In fact, Calvo et al. reported that in mammary epithelial cells, high-stiffness matrices promote the cytoplasmic retention of YAP [27], which aligns with our observations in adipocytes.

Moreover, increasing the concentration of hydrogels not only alters the matrix stiffness but may also affect cell–matrix interactions and nutrient diffusion [28]. These factors, working together, could modulate YAP activity and influence the adipocyte differentiation process through multiple signaling pathways. Rho GTPases and the actin cytoskeleton are crucial in regulating YAP activity through mechanical forces [29], suggesting a potential direction for future research.

Our experimental results demonstrate that the concentration of hydrogels significantly affects the intracellular localization and activity of YAP, thereby regulating the differentiation process of adipocytes. Through fluorescence co-localization analysis, we assessed the localization of YAP in the nucleus and its overlap with nuclear regions. Increasing the hydrogel concentration from 0.5 mg/mL to 2 mg/mL significantly altered the shuttling behavior of YAP between the nucleus and cytoplasm (Figure 3A–C). The 0.5 mg/mL hydrogel exhibited the highest YAP nuclear localization, with YAP predominantly translocating to the nucleus in this softer microenvironment. The overlap rate between YAP and the nucleus was 69.44% ± 13.08%. As the hydrogel concentration increased, YAP’s nuclear localization gradually decreased. In the 1 mg/mL hydrogel, the overlap rate between YAP and the nucleus was 60.24% ± 10.6%, while in the 2 mg/mL hydrogel, this value dropped to 34.54% ± 11.08%, indicating that the activation of YAP nuclear localization is negatively correlated with the hydrogel concentration. At higher hydrogel concentrations, YAP remains largely inactive, predominantly residing in the cytoplasm (Figure 3C). In the cytoplasm, YAP fluorescence intensity significantly increased with the stiffness of the hydrogels, suggesting that YAP was unable to translocate to the nucleus. The cytoplasmic retention of YAP in hydrogels was 30.56% ± 9.48% for 0.5 mg/mL, 39.8% ± 8.84% for 1 mg/mL, and 65.46% ± 9.48% for 2 mg/mL. These data further demonstrate that YAP activation is dependent on the mechanical stiffness of the hydrogels.

In the fluorescence co-localization experiment (Figure 3D,E), we focused on analyzing the nuclear distribution of YAP in hydrogels of varying concentrations. The results showed that as the hydrogel concentration increased, the distribution of YAP within the nucleus gradually decreased, indirectly supporting the conclusion that YAP nuclear translocation is regulated by the stiffness of the hydrogels. Notably, after YAP enters the nucleus, it typically cooperates with transcription factors to regulate gene expression, thereby influencing cell proliferation and differentiation. Conversely, when YAP predominantly remains in the cytoplasm, its retention is often associated with the promotion of mature adipocyte differentiation. To further verify whether hydrogel concentration is a key factor influencing YAP nuclear translocation, we performed a Western blot analysis, which corroborated these findings. We measured the expression of YAP protein in hydrogels of different concentrations (Figure 3F–I). The results indicated that with increasing hydrogel concentration, YAP expression in the nucleus significantly decreased, while its expression in the cytoplasm increased accordingly. This aligns with prior findings on YAP nuclear translocation, indicating that hydrogel concentration influences both YAP intracellular localization and its protein expression. YAP, a crucial effector in the Hippo signaling pathway, is pivotal in mechanotransduction and cell fate determination [30]. Previous research indicates that the nuclear localization of YAP, an active form, enhances stem cell proliferation and suppresses differentiation [31]. Our experimental findings further reveal the significant impact of hydrogel concentration on the intracellular localization and activity of YAP. YAP progressively translocated from the nucleus to the cytoplasm as the hydrogel concentration increased from 0.5 mg/mL to 2 mg/mL. In the low-concentration hydrogel (0.5 mg/mL), YAP primarily accumulated in the nucleus, whereas at higher concentrations (2 mg/mL), YAP was predominantly retained in the cytoplasm. YAP nuclear localization and protein expression levels decreased as hydrogel concentration increased, highlighting the significant role of hydrogel mechanical stiffness in regulating YAP nuclear translocation and adipocyte differentiation. This phenomenon was confirmed through fluorescence co-localization experiments and Western blot protein expression analyses, further supporting the regulatory effect of hydrogel stiffness on YAP activity and adipocyte differentiation.

### 3.8. Inhibiting YAP Promotes Adipogenesis

YAP is an important transcriptional coactivator that controls the process of differentiation into adipocytes. Indeed, various publications have documented that the inhibition of YAP promotes the differentiation of adipocytes. It has been indicated that dysfunctional adipocytes trigger YAP/TAZ signaling through a process that leads to dedifferentiation and tumor development. Inhibition of YAP/TAZ in these models restored the function of the adipocytes and diminished inflammation, thus suggesting that targeting YAP improves adipocyte differentiation [31]. Further research indicates that YAP repression promotes the differentiation of other cell types, including endothelial cells, by inducing key transcription factors such as FLI1 [32]. These findings suggest that YAP repression may play a broader role in promoting the differentiation of multiple cell types into adipocytes [33]. YAP, in relation to myxoid liposarcoma, represses adipogenic differentiation; hence, inhibition of YAP should reverse this effect, promoting normal adipocyte differentiation [34].

Previous research indicates that YAP signaling regulates lipid droplet accumulation, with expression levels directly affected by matrix stiffness. We inhibited YAP signaling to investigate its impact on lipid droplet accumulation in hydrogel-induced adipogenesis. After inhibiting YAP signaling for 48 h, we stained the lipid droplets with BODIPY and analyzed the fluorescence intensity using ImageJ 1.52P software. The results, as depicted in Figure 4A, indicated that lipid droplet accumulation in the control group was consistent with the observations in Figure 2A. Inhibition of YAP signaling led to increased lipid droplet accumulation in the inhibited group’s hydrogel compared to the control group. Statistical analysis of Figure 4B reveals that the inhibition group exhibited higher fluorescence intensity compared to the control group, suggesting that inhibiting YAP signaling, which retains YAP in the cytoplasm, enhances lipid droplet accumulation. This was corroborated by BODIPY staining of lipid droplets in various hydrogels, confirming that YAP signaling pathway inhibition promotes lipid droplet accumulation. As shown in Figure 4C–E, Western blot experiments also corroborate this finding. In the experiments, inhibiting the YAP signaling pathway with VP led to increased protein expression compared to the non-inhibited pathway. Matrix stiffness impacts YAP expression, which subsequently influences lipid droplet accumulation, highlighting YAP’s essential role in adipogenesis.

### 3.9. Hydrogel Concentration Regulates Adipocyte Migration Behavior

Hydrogel concentration significantly influences adipocyte migration and behavior, particularly through its mechanical properties and matrix density. Research indicates that the stiffness and density of hydrogels can regulate the phenotype and migration dynamics of adipocytes. Mechanical characteristics, such as stiffness, influence adipocyte migration, with softer hydrogels often promoting different migration strategies compared to stiffer ones. This variation significantly impacts the overall behavior of adipocytes [35].

The co-culture system validated the impact of hydrogel concentration on adipocyte differentiation and migration, revealing changes in YAP activity and migration patterns in hydrogels at 0.5 mg/mL, 1 mg/mL, and 2 mg/mL. Hydrogel concentration clearly had an impact on intracellular localizations and activity of YAP, further influencing the differentiation and migratory capacity of adipocytes. The differentiation and migration of adipocytes in hydrogels of different concentrations were studied in the co-culture system, in which an upper chamber with 8 µm pores was placed in a lower 24-well plate (Figure 5B). The observations at the bottom of the 24-well plate after 9 days of differentiation, as shown in (Figure 5C), indicate that under the 2 mg/mL hydrogel conditions, a few cells were able to migrate to the bottom of the plate. In contrast, no cell migration was observed under the 0.5 mg/mL and 1 mg/mL conditions. Figure 5D presents quantitative data on cell migration. Figure 5A demonstrates cell migration across all tested hydrogel concentrations at the bottom of the 8 µm pore chamber, with a significant increase in migrating cell numbers corresponding to higher hydrogel concentrations. The 2 mg/mL hydrogel condition yielded the highest number of migrating cells.

These findings point out that hydrogel concentration, or matrix stiffness, has a great influence on the differentiation and migration of adipocytes. Previous studies indicate that the matrix’s physical properties, particularly stiffness, significantly affect cellular behaviors like differentiation and migration [36]. Our findings align with previous studies and emphasize the critical role of matrix stiffness in adipocyte differentiation. Moreover, we found that even when the cells could migrate through the 8 µm pores, they only reached the bottom of the 24-well plate under the condition of 2 mg/mL. This indicated that matrix stiffness might influence not only the tendency for the cell to migrate but also their ability and persistence.

### 3.10. Hydrogel Stiffness Modulates YAP-Mediated Adipocyte Differentiation

During adipocyte differentiation, the cells are influenced by mechanical forces from hydrogels of varying hardness, and YAP exhibits differential sensitivity to these forces. Mechanical force enables YAP to move between the nucleus and cytoplasm. The nuclear–cytoplasmic translocation of YAP and its phosphorylation are beneficial for the formation of mature adipocytes. Figure 6 illustrates that in softer hydrogels, YAP less frequently shuttles from the nucleus to the cytoplasm, leading to reduced phosphorylation and decreased lipogenesis. In the harder hydrogel, increased YAP shuttling from the nucleus to the cytoplasm and subsequent phosphorylation result in higher lipogenesis compared to the softer hydrogel. In summary, stiffer hydrogels are more conducive to fat differentiation.

## 4. Discussion

This study has systematically investigated how collagen hydrogels of different stiffnesses affect the differentiation of ADSCs and then disclosed the regulatory function of YAP signaling in such a process. The increase in hydrogel stiffness significantly enhanced the adipogenic differentiation capability of adipose-derived stem cells (ADSCs), as evidenced by a greater number and size of lipid droplets, along with markedly higher mRNA expression of adipocyte-specific genes such as ADIPOQ, LPL, PPAR, and FABP4. The 2 mg/mL hydrogel showed increased cytoplasmic YAP phosphorylation and reduced nuclear YAP expression compared to the 0.5 mg/mL and 1 mg/mL hydrogels, suggesting that stiffer ECM retains YAP in the cytoplasm, thereby enhancing adipogenesis.

The previous classical literature has clearly elucidated the pivotal role of YAP in adipogenesis. To prevent mesenchymal cells from drifting toward adipocytes in orthopedic and reconstructive surgery, Oliver-De La Cruz et al. utilized bioengineering tools that distinguish between mechanical and biological stimuli to explore the inhibition of adipogenesis. Their research underscores the critical importance of YAP phosphorylation in the adipogenesis of mesenchymal stem cells [37]. In another study, Lorthongpanich et al., confirmed that YAP is a key regulator in the differentiation of mesenchymal stem cells into osteocytes and adipocytes in a 2D environment, using pharmacological and genetic experimental techniques [32]. Leptin, a hormone secreted by adipocytes, regulates energy–metabolism balance and is closely associated with feeding behavior in animals. Choi et al. employed gene manipulation techniques to knock out the upstream regulatory molecules Lats1 and Lats2 of the Hippo signaling pathway, resulting in the dedifferentiation of adipocytes and an increase in leptin levels. They further confirmed that YAP regulates the leptin gene, playing an important role in maintaining energy metabolism balance [38]. Pan et al. also used gene manipulation techniques to knock out YAP in osteoblasts, leading to the inhibition of bone differentiation and an increase in adipogenesis. However, maintaining bone homeostasis relies not only on the YAP/TAZ signaling pathway but also on the integration of the β-catenin signaling pathway [39].

The commonality among these studies lies in their use of fundamental experimental techniques to conduct foundational research on the relationship between YAP, adipogenesis, and energy metabolism. In contrast, our study applies tissue engineering principles to mimic the microenvironment of adipose tissue, specifically examining how substrate stiffness influences adipogenesis through YAP signaling, with the ultimate goal of serving clinical purposes.

Several studies indicate that during the process of mechanotransduction, YAP activity is regarded as a key factor, as its nucleocytoplasmic shuttling is influenced by ECM stiffness [40,41]. These findings indicate that inhibiting YAP signaling significantly increases lipid droplet formation, supporting the hypothesis of a positive correlation between YAP inhibition and adipogenic differentiation. Stiffer hydrogels promote YAP phosphorylation, forcing it to remain cytoplasmic, which in turn reduces its inhibition of gene transcription and thereby enhances adipocyte differentiation. This is in agreement with the mechanosensitive properties of YAP previously reported on 2D substrates [42,43,44,45,46] and suggests that YAP remains a central regulator of both mechanosensing and differentiation on 3D substrates as well [23,47,48].

Although this paper has illustrated a mechanism of how the ECM stiffness may influence adipogenic differentiation, there are several limitations that have to be considered. Although our hydrogel model effectively simulates various ECM stiffness levels found in the adipose tissue microenvironment, it does not capture the complete complexity of the in vivo microenvironment, such as the effects of other cytokines and biochemical signals. The YAP signaling pathway is a primary focus in regulating adipogenic differentiation, whereas the roles of other mechanotransduction pathways, such as TGF-β and Wnt/β-catenin, remain poorly characterized.

Future research could further investigate how ECM composition, stiffness, and other signaling molecules interact under different physiological conditions to influence adipogenesis. Additionally, exploring methods to precisely regulate adipose tissue formation and differentiation by modulating ECM’s physical properties and the YAP signaling pathway using in vivo models of adipogenesis may provide novel strategies for tissue engineering and the treatment of obesity-related diseases.

## 5. Conclusions

This study systematically examines how extracellular matrix stiffness affects ADSCs differentiation, with a focus on the YAP signaling pathway. Using different concentrations of collagen hydrogels, which allow for a good imitation of the in vivo microenvironment, we determined that the degree of adipocyte differentiation positively correlated with increased stiffness of the hydrogels. Moreover, the inhibition of YAP signaling promoted the accumulation of lipid droplets, reinforcing the idea that ECM stiffness modulates adipogenesis by altering YAP’s mechanosensitive properties and its phosphorylation state. Noticeably, in 2 mg/mL hydrogels, larger pores and higher elastic modulus strongly supported the differentiation of ADSCs. These findings highlight the regulatory role of YAP in stiffness-dependent adipogenesis, offering a novel theoretical foundation for designing tissue engineering materials and guiding therapeutic strategies for obesity-related diseases.

## Figures and Tables

**Figure 1 cells-13-01905-f001:**
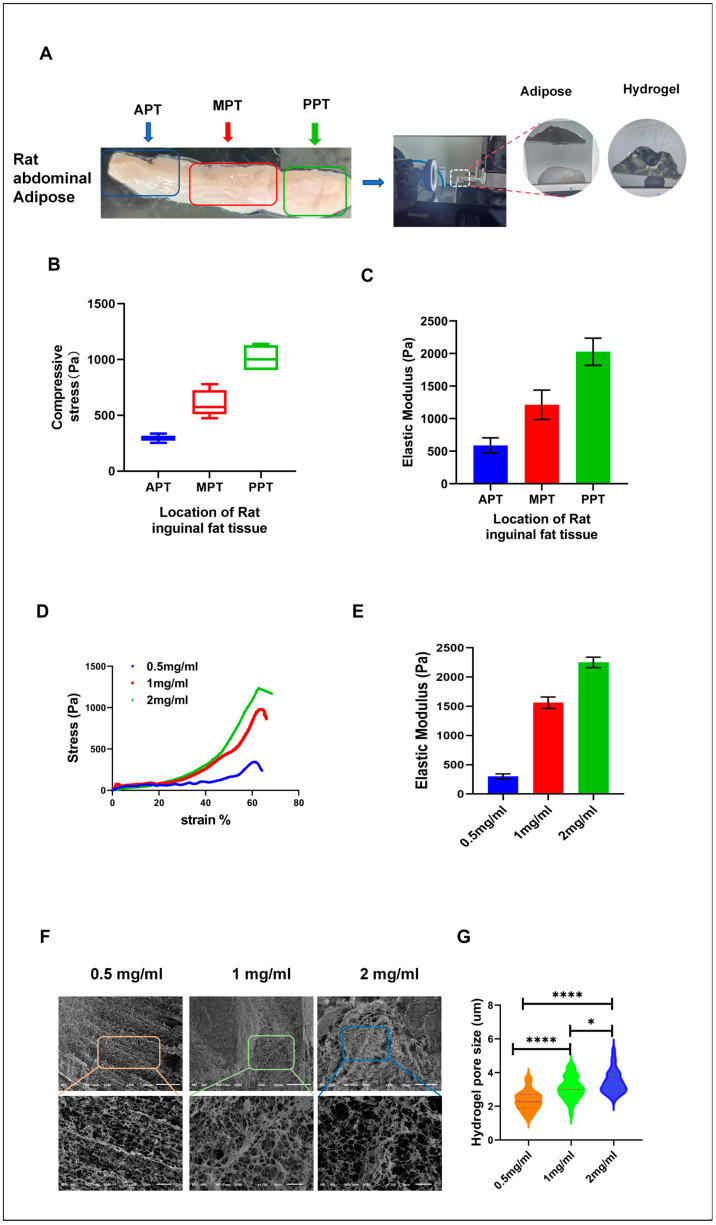
Mechanical properties of adipose tissue and physical characteristics of hydrogels at different concentrations. (**A**) Results of stress and elastic modulus tests on different regions of adipose tissue. (**B**) Statistical results of stress testing in adipose tissue. (**C**) Testing results of the elastic modulus in various regions of adipose tissue (n = 5). (**D**) Stress testing results of hydrogels at different concentrations. (**E**) Elastic modulus testing results of hydrogels at varying concentrations (n = 5). (**F**) SEM images of hydrogels at different concentrations. (**G**) Statistical analysis of pore sizes in hydrogels at different concentrations (n = 80). The pore diameters were analyzed using Nano Measurer 1.2 software and statistically processed using GraphPad Prism 8.0.2 software.* *p* < 0.05. **** *p* < 0.0001.

**Figure 2 cells-13-01905-f002:**
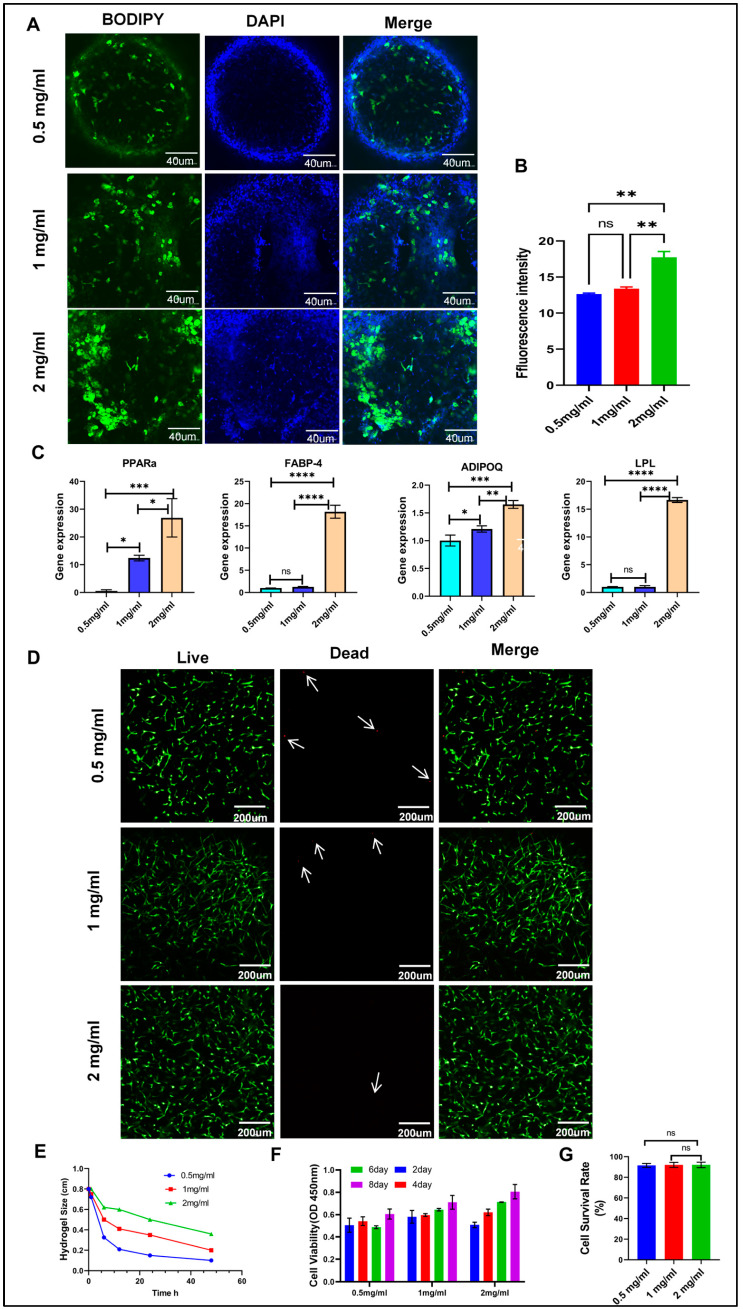
Comprehensive assessment of the effects of hydrogels on adipocyte function and material mechanical properties. (**A**) Observation of adipocyte differentiation in hydrogels via fluorescence staining. (**B**) Quantitative analysis of fluorescence signals related to adipocyte differentiation. (**C**) Measurement of the expression levels of specific genes associated with adipocyte differentiation. (**D**) Evaluation of adipocyte viability within the hydrogels. (**E**) Assessment of the impact of cell embedding on hydrogel contraction (n = 5). (**F**) Monitoring of cell proliferation in hydrogels over different culture time periods. (**G**) Statistical analysis of cell viability data. Fluorescence intensity was analyzed using ImageJ 1.52P software and statistically processed using GraphPad Prism 8.0.2 software. * *p* < 0.05. ** *p* < 0.01. *** *p* < 0.001. **** *p* < 0.0001. ns: not significant (n = 4).

**Figure 3 cells-13-01905-f003:**
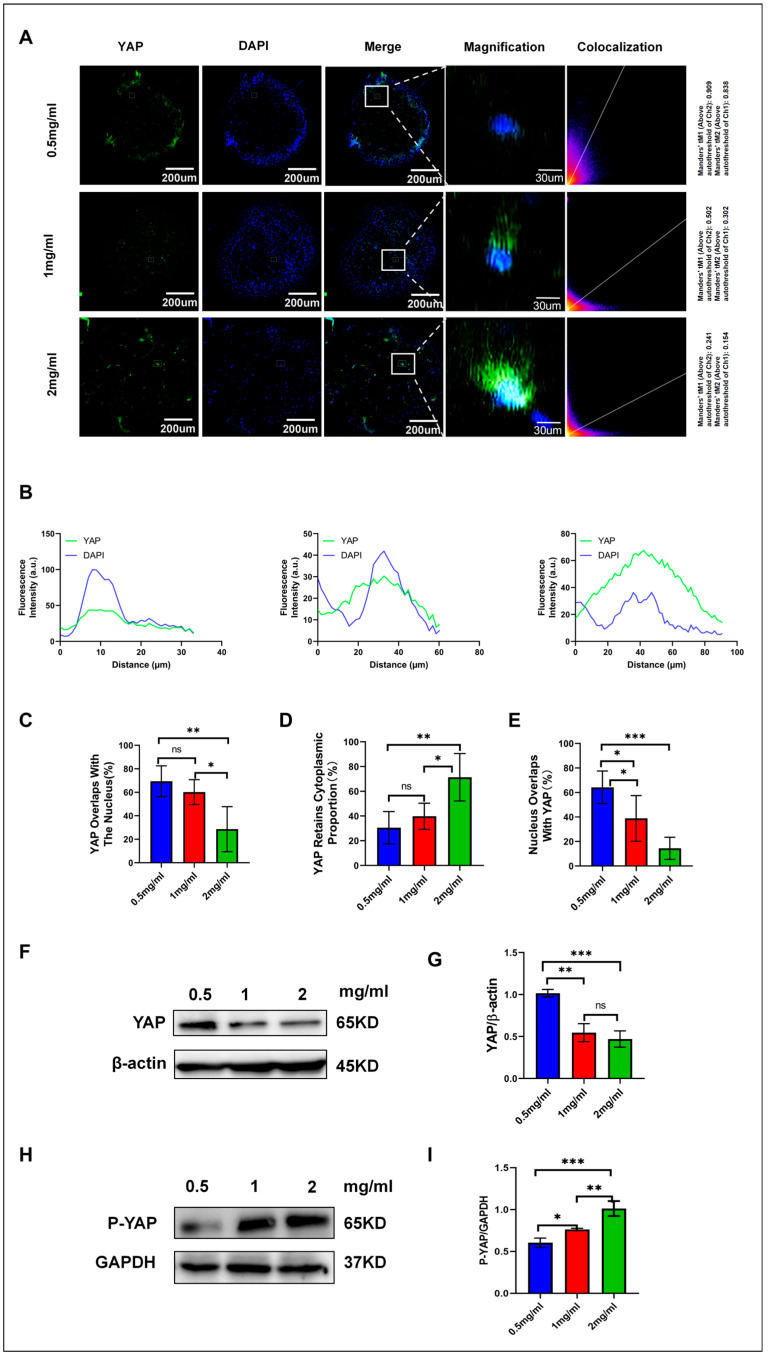
Immunofluorescence co-localization and protein analysis of YAP expression in hydrogels. (**A**) Immunofluorescence staining of YAP in adipocytes cultured in hydrogels of different concentrations. YAP is shown in green, and the nucleus is stained with DAPI (blue), with the merged images displaying co-localization. In the histogram, the red represents YAP signals (green fluorescence channel), while the blue represents DAPI signals (nuclei). (**B**) A single cell is outlined as the test subject, as shown in (**A**). The nucleus is labeled in blue and YAP is labeled in green, outlined by a white line. The fluorescence intensity distribution of YAP and DAPI along the white box is analyzed, with the intensity values normalized to the average YAP intensity. (**C**) Percentage of YAP co-localized with the nucleus. (**D**) Proportion of YAP signals not overlapping with the nucleus. (**E**) Percentage of YAP localized within the nucleus. (**F**) Western blot analysis of YAP protein expression in varying hydrogel concentrations. (**G**) Quantitative analysis of YAP protein expression across varying hydrogel concentrations. (**H**) Western blot analysis of phosphorylated YAP (P-YAP) protein expression in hydrogels of different concentrations. (**I**) Quantitative analysis of P-YAP protein expression across various hydrogel concentrations was conducted. Fluorescence intensity and co-localization data were processed with ImageJ 1.52P software, and statistical analysis was executed using GraphPad Prism 8.0.2 software. Statistical significance is indicated as * *p* < 0.05. ** *p* < 0.01, *** *p* < 0.001, and ns for not significant (n = 3).

**Figure 4 cells-13-01905-f004:**
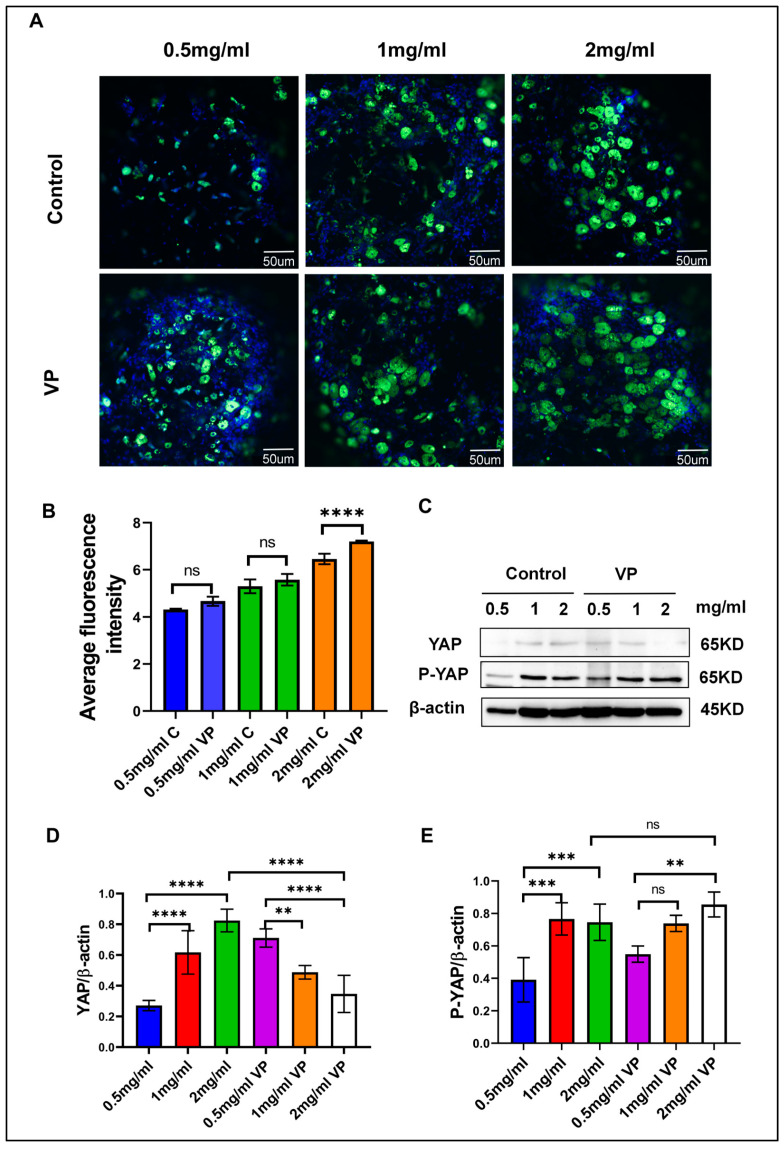
Inhibition of the YAP signaling pathway promotes adipogenesis. (**A**) BODIPY fluorescence staining of lipid droplets following inhibition of the YAP signaling pathway using the VP inhibitor. (**B**) Fluorescence intensity analysis of lipid droplets in hydrogels of different concentrations. (**C**) Changes in YAP/P-YAP protein levels after VP inhibition. (**D**,**E**) Quantitative analysis of protein expression changes after YAP signaling inhibition. The fluorescence intensity in Figures (**A**) and (**C**) was analyzed using ImageJ 1.52P software, and statistical data were processed using GraphPad Prism 8.0.2 software. Statistical significance is denoted as ** *p* < 0.01, *** *p* < 0.001, **** *p* < 0.0001, and ns for non-significant differences (n = 3).

**Figure 5 cells-13-01905-f005:**
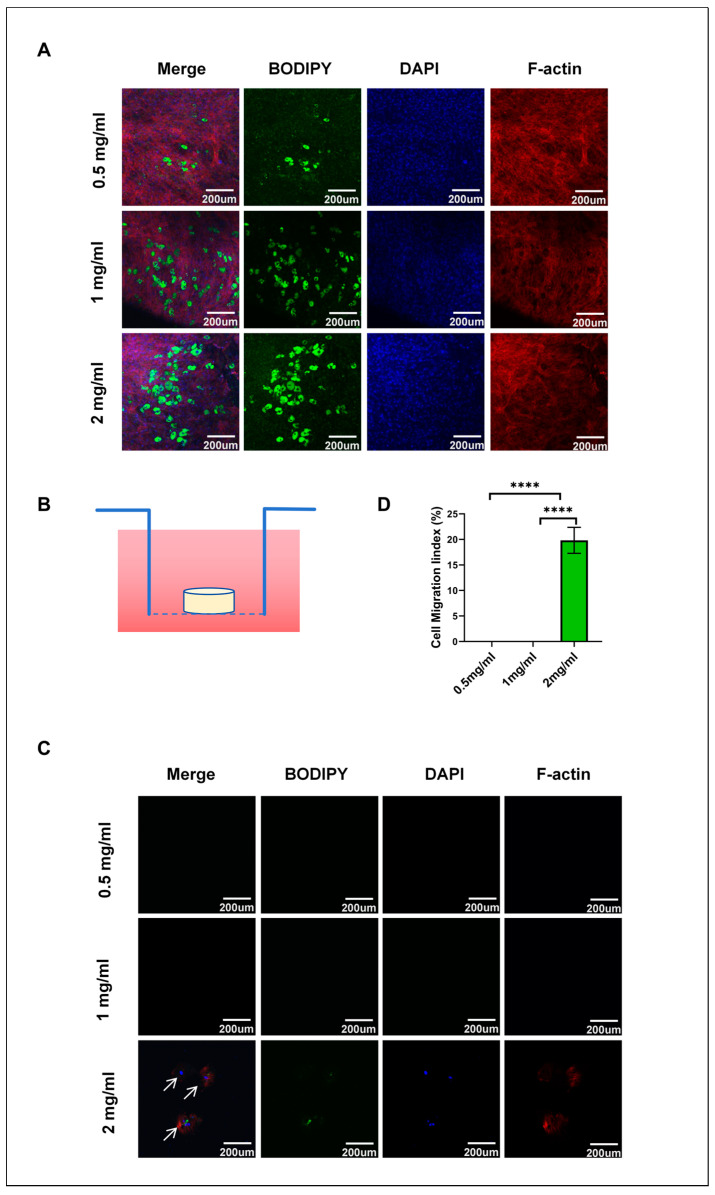
Effect of hydrogel stiffness on adipocyte migration. (**A**) Cell staining at the bottom of the 8 µm pore size chamber, showing adipocyte migration characteristics in hydrogels of different concentrations. (**B**) Schematic diagram of the co-culture system featuring an 8 µm pore size upper chamber and a lower 24-well plate for observing cell migration behavior. (**C**) Cell migration in the bottom of the 24-well plate, with lipid droplets stained green using BODIPY and actin filaments stained red using fluorescence staining. (**D**) Quantitative analysis of cell migration. Fluorescence intensity was analyzed using ImageJ 1.52P software and statistically processed using GraphPad Prism 8.0.2 software. **** *p* < 0.0001. (n = 3).

**Figure 6 cells-13-01905-f006:**
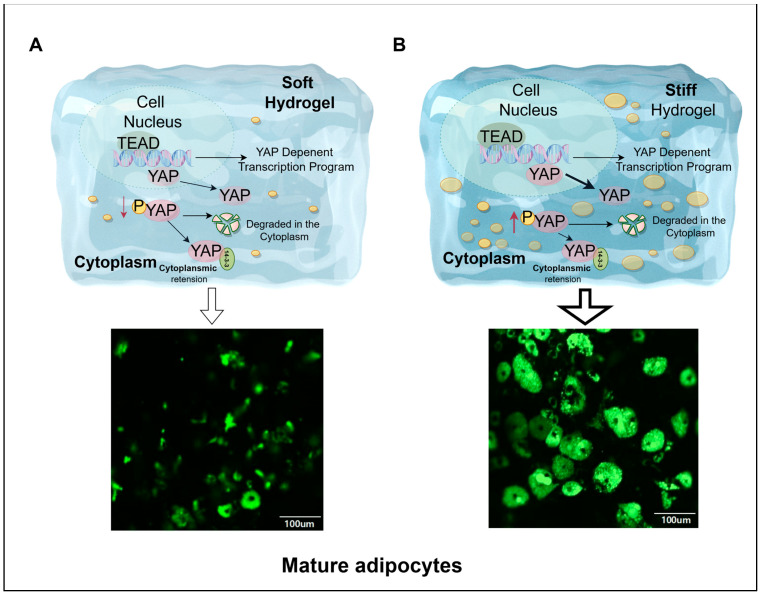
Adipocyte differentiation is regulated by the YAP signaling pathway. (**A**,**B**) show collagen hydrogels of different stiffness. In softer hydrogels (**A**), the translocation of YAP from the nucleus to the cytoplasm is diminished, along with reduced phosphorylation, resulting in decreased lipid droplet accumulation. In contrast, in stiffer hydrogels (**B**), YAP translocates more readily from the cytoplasm to the nucleus, accompanied by elevated phosphorylation, which enhances lipid droplet accumulation. Hydrogels with varying matrix stiffness influence YAP’s relocalization between the nucleus and cytoplasm under mechanical stress. When YAP remains in the cytoplasm, it undergoes phosphorylation, promoting the initiation of the adipocyte differentiation program. Therefore, matrix stiffness ultimately affects adipocyte differentiation by regulating both the cellular localization and activity state of YAP. This figure was created with Figdraw (HOME for Researchers).

**Table 1 cells-13-01905-t001:** RT-PCR Primer.

Target	Forward Primer (3′–5′)	Reverse Primer (5′–3′)
**GAPDH**	**GCATCTTCTTGTGCAGTGCC**	**GATGGTGATGGGTTTCCCGT**
**LPL**	**GAGAAGGGGCTTGGAGATGT**	**ATGCCTTGCTGGGGTTTTCT**
**ADIPOQ**	**CCGTTCTCTTCACCTACGAC**	**TTCCCCATACACTTGGAGCC**
**PPARa**	**TCGTGGAGTCCTGGAACTGA**	**CTTCAGTCTTGGCTCGCCTC**
**FABP-4**	**AGAAGTGGGAGTTGGCTTCG**	**ACTCTCTGACCGGATGACGA**

**Table 2 cells-13-01905-t002:** Antibody statistics.

Antibody	Company	Item Number	Attributes	Dilution Ratio
**Phospho-YAP**	**Cell Signaling**	**4911**	**Rabbit**	**1:1000**
**YAP**	**Cell Signaling**	**14074S**	**Rabbit**	**1:1000**
**GAPDH**	**Santa Cruz Biotechnology**	**SC-47724**	**Mouse**	**1:500**

## Data Availability

The datasets generated and analyzed during this study are available from the corresponding author upon reasonable request.

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
