# Peer review of "YAP and ECM Stiffness: Key Drivers of Adipocyte Differentiation and Lipid Accumulation"

_cells, 2024, doi:10.3390/cells13221905_

Round 1

Reviewer 1 Report

Comments and Suggestions for Authors

The authors have done a wonderful job of compiling a manuscript where-in they correlate the matrix strength with sections of Rat adipose tissue stiffness, further clarifying the relation of YAP in adipose tissue regeneration.  This is more of a model system to study in-situ tissue regeneration.

Altough they have to explain in detail how they think that this scaffold material can be used therapeutically as they claim in the discussions. If they are unable to they should just claim to have a model to study adipose tissue regeneration and YAP involvement therein. 

Figures: 

1. Figure legends must include the N wherever possible to indicate how statistical analysis was performed

2. Figure 2 a - What looks like images require rearrangement since the image magnification correlate with concentration , it should ideally be the rows have been misplaced with columns. 

Minor comments:

 1. References must be uniform, at some places there are titles (ref 28)in capitals while at other placed all authors are in abbreviaitons (ref 4)

Author Response

Reviewer #2:

A: We agree with the comments from reviewer. In next study, we will use subcutaneous transplantation model in animals to investigate the effects of hydrogels with different stiffness on  fat formation and detect the nuclear-cytoplasmic translocation of the key signaling molecular  YAP during this process, providing reliable experimental data for the advancement of aesthetic medicine.

Figures:

Q1: Figure legends must include the N wherever possible to indicate how statistical analysis was performed

A: We have made supplements according to the reviewer's comments and highlighted in blue color.

Q2: Figure 2 a - What looks like images require rearrangement since the image magnification correlate with concentration, it should ideally be the rows have been misplaced with columns.

A: Varying concentrations of collagen, along with the presence of cells, can result in a shrinkage phenomenon, often creating the illusion of inconsistent magnification. However, the magnification of all images is consistent.

Minor comments:

 Q3: 1. References must be uniform, at some places there are titles (ref 28)in capitals while at other placed all authors are in abbreviaitons (ref 4)

A: We have made corrections according to the reviewer's comments and highlighted in blue  color.

Reviewer 2 Report

Comments and Suggestions for Authors

The manuscript entitled “YAP and ECM Stiffness: Key Drivers of Adipocyte Differentiation and Lipid Accumulation” investigates the dependence between ECM stiffness and ADSC differentiation. In addition, this work is focused on YAP as a key regulator for adipogenesis and lipid droplet accumulation. The authors present detailed and coherent research, however, there are some concerns that need to be addressed before further consideration for publication in this journal.

  1. The authors should better clarify what is the novelty and originality of this work as compared to previous studies reported both the relation between ECM stiffness and adipogenesis and the key role of YAP modulating this process  (https://doi.org/10.1016/j.biomaterials.2019.03.009;https://doi.org/10.1186/s13287-019-1494-4;https://doi.org/10.1038/s42255-024-01045-4;https://doi.org/10.1038/s41413-018-0018-7; ).
  2. In Figure 1 and in the methods section – for the evaluation of the mechanical properties, the authors should include what is the sample size, what is the collagen hydrogel volume, what is the cell density (cells/ml), and the authors should perform the mechanical tests on ADSC cellularized hydrogel plugs which may affect the hydrogel stiffens (for several time points post cell seeding). In addition, it is not clear which graph slope was taken for the Young’s modulus calculation (for which strain [%] values).
  3. Following the previous comment, it seems that the softer the hydrogel is, a significant shrinkage level of the plugs is observed. This is a known phenomenon developed by the contraction forces extracted by the cells and can lead to an ischemic core of the plug and lower viability and functionality of the cells. It may affect the results thus for validating the results the authors should consider performing another experiment with hydrogel-coated slides/wells to prevent the shrinkage effect and assess the ECM stiffness-adipogenesis relation.
  4. The quality of the images should be addressed in all figures, the authors should add noticeable scale bars in all images, also should improve resolution and contrast in all the microscope images.
  5. The authors should describe in the methods which statistical tests were conducted in this work.
  6. For the viability test in Figure 2D it is very hard to see the red-stained dead cells, please improve the quality of these images (and include scale bars). In addition, the authors should quantify the viability rate [% of living cells] in a graph.
  7. For all the YAP-related images (for example figure 3A) the authors should add higher magnification that enables the reader to observe easily single cell nuclei. It is important to indicate the translocation of YAP in the nuclei and the level of the YAP in the cytoplasm (x100 magnification is recommended).    
  8. For the YAP inhibitor experiments presented in Figure 4, the authors should assess the plug size (diameter), shrinkage level, and cell viability to better understand the effect of the inhibitor on cell functionality and the direct and indirect effects on the differentiation rate.
  9. Figure 5C images are in low quality, it is difficult to notice the cells please change to higher quality images and add a quantification for the migration rate.
  10. In Figure 6 the illustrations present the same mechanism it is not clear how the mechanism is differentiated between the two states. The authors should better clarify what are the govern elements that differentiate the two states.

Comments on the Quality of English Language

Should conduct proof reading

Author Response

Reviewer #1:

Q1: The authors should better clarify what is the novelty and originality of this work as compared to previous studies reported both the relation between ECM stiffness and adipogenesis and the key role of YAP modulating this process  (https://doi.org/10.1016/j.biomaterials.2019.03.009;https://doi.org/10.1186/s13287-019-1494-4;https://doi.org/10.1038/s42255-024-01045-4;https://doi.org/10.1038/s41413-018-0018-7; ).

A: Thank you very much to the reviewer for the excellent literature provided, which has enhanced our understanding of YAP's role in adipogenesis. In line with the reviewer’s suggestions, we conducted a comparative analysis in the discussion section on page 30-31, highlighting the key points in red.

Q2: In Figure 1 and in the methods section – for the evaluation of the mechanical properties, the authors should include what is the sample size, what is the collagen hydrogel volume, what is the cell density (cells/ml), and the authors should perform the mechanical tests on ADSC cellularized hydrogel plugs which may affect the hydrogel stiffens (for several time points post cell seeding). In addition, it is not clear which graph slope was taken for the Young’s modulus calculation (for which strain [%] values).

A: hydrogel volume is 100uL, cell density is 1.5×105 cells/mL. Highlight in red color on page 5.

The overall strain-stress curves of the three hydrogels is nonlinear. However, the strain range of 40% to 50% is approximately linear. Thus, this specific range was chosen for analysis to determine the Young’s modulus.

Q3: Following the previous comment, it seems that the softer the hydrogel is, a significant shrinkage level of the plugs is observed. This is a known phenomenon developed by the contraction forces extracted by the cells and can lead to an ischemic core of the plug and lower viability and functionality of the cells. It may affect the results thus for validating the results the authors should consider performing another experiment with hydrogel-coated slides/wells to prevent the shrinkage effect and assess the ECM stiffness-adipogenesis relation.

A: To address the reviewer’s suggestions for additional experiments, we requested an extension from the editorial office, which was approved. Please refer to the supplementary data S5 for the results of the additional experiments. We made every effort to control the retraction of the hydrogel as requested by the reviewer while observing its effects on adipogenic differentiation. Although the 0.5 mg/ml and 1 mg/ml groups showed slight improvements in retraction, the differences compared to the 2 mg/ml group remained quite significant. In terms of adipogenic differentiation, the 2 mg/ml group still outperformed the other two groups.

Q4: The quality of the images should be addressed in all figures, the authors should add noticeable scale bars in all images, also should improve resolution and contrast in all the microscope images.

A: We agree with the reviewer‘s comments and have made improvements to all the images, including the scale. In particular, for Figure 2A, we have replaced it with a clearer image taken at a higher magnification and provided a new quantitative analysis (Figure 2B).

Q5: The authors should describe in the methods which statistical tests were conducted in this work.

A: “Statistical Analysis” is put in the methods according to the reviewer's comments and highlighted in red color.

Q6: For the viability test in Figure 2D it is very hard to see the red-stained dead cells, please improve the quality of these images (and include scale bars). In addition, the authors should quantify the viability rate [% of living cells] in a graph.

A: We followed the reviewer’s instructions and conducted cell viability staining again. The results were consistent with our previous findings, indicating that our 3D hydrogel is very beneficial for cell growth and survival. Quantify the viability rate have been added to Figure 2G For the results of the supplementary experiments on cell viability, please refer to the supplementary data S8。Previously, Figures 2D and 2G showed cell viability at 24 hours. To gain a more comprehensive understanding of cell viability in hydrogel-embedded conditions, we captured live/dead cell data at various time points and performed statistical analysis. The results indicate that dead cells are difficult to detect in hydrogels of varying concentrations, further highlighting the effectiveness of the hydrogels we developed in promoting cell survival.

Q7: For all the YAP-related images (for example figure 3A) the authors should add higher magnification that enables the reader to observe easily single cell nuclei. It is important to indicate the translocation of YAP in the nuclei and the level of the YAP in the cytoplasm (x100 magnification is recommended).

A: We agree with the reviewer's suggestions and have made the corresponding additions. Please refer to Figure 3A (magnification), which shows a localized magnified view of the immunofluorescence colocalization.

Q8: For the YAP inhibitor experiments presented in Figure 4, the authors should assess the plug size (diameter), shrinkage level, and cell viability to better understand the effect of the inhibitor on cell functionality and the direct and indirect effects on the differentiation rate.

A: We agree with the comments from reviewer. Please refer to the supplementary data S7 for the results of the additional experiments.

Q9: Figure 5C images are in low quality, it is difficult to notice the cells please change to higher quality images and add a quantification for the migration rate.

A: We agree with the reviewer's suggestions and have made the corresponding additions. Please refer to Figure 5D.

Q10: In Figure 6 the illustrations present the same mechanism it is not clear how the mechanism is differentiated between the two states. The authors should better clarify what are the govern elements that differentiate the two states.

A: We agree with the comments from reviewer. In line with the reviewer’s suggestions,  we have provided a clearer explanation on pages 29-30 and included annotations in Figure 6 and highlighted in red color.

Round 2

Reviewer 2 Report

Comments and Suggestions for Authors

The authors have addressed most of my comments. I just have a minor comment regarding Figure 6 – the mechanism illustration is still unclear. Maybe the authors should emphasize better the reduced phosphorylation by changing the arrow direction (pointing down) in A and adding an illustration of lipid droplet formation in B. 

Author Response

Q1. The authors have addressed most of my comments. I just have a minor comment regarding Figure 6 – the mechanism illustration is still unclear. Maybe the authors should emphasize better the reduced phosphorylation by changing the arrow direction (pointing down) in A and adding an illustration of lipid droplet formation in B. 

  1. We agree with the reviewer’s suggestions and have made the corresponding modifications and additions. Please refer to Figure 6.
